# Proposal of a Multi-Criteria Model for the Evaluation of Territorial Development Plans: An Application in Chile’s Lagging Areas

**DOI:** 10.3390/ijerph191811312

**Published:** 2022-09-08

**Authors:** Sara Arancibia-Carvajal, Felipe Petit-Laurent, María Paz Troncoso, Manuel Vargas-Vargas

**Affiliations:** 1Institute of Basic Sciences, Faculty of Engineering and Sciences, University Diego Portales, Santiago 8370191, Chile; 2Huella Local Foundation, Santiago 8420200, Chile; 3Director of Development, Faculty of Economics and Business, Universidad San Sebastián, Santiago 8420524, Chile; 4Faculty of Economic and Business Sciences, University of Castilla de la Mancha, 02006 Albacete, Spain

**Keywords:** regional inequalities, regional and local development, territorial planning, vulnerable communities, Chile

## Abstract

The evaluation of Territorial Development Plans (TDP) is a challenge most Latin American countries face. The problem arises in establishing a model to evaluate TDP that meets the criteria and indicators established in a national policy or regulatory framework under local needs. This study proposes an application to evaluate the TDP of the lagging areas in Chile based on the AHP multi-criteria methodology. This methodology allows to objectively unify the evaluation of the different plans, combining the different dimensions, objectives, scales, and judgments of the experts present in the evaluation process without sacrificing the quality, reliability, and participation of the actors involved. The model is flexible to changes in the criteria, as it can be updated according to the needs over time. An efficient and effective tool is provided to support decision-makers in formulating better development plans to bridge the gaps in territorial groups with high vulnerability.

## 1. Introduction

Today, one of the most relevant challenges of any developing society is to ensure that development is achieved harmoniously throughout its territory. To this end, there has been broad agreement on the need to recognize and respond to local and regional needs, creating an appropriate and flexible combination of decentralization and centralization [1]. Decentralization refers to the devolution of responsibilities, political decisions, and fiscal powers to subnational, regional, and local levels of government. Effective decentralization requires that several conditions be met, including effective coordination mechanisms and monitoring systems [2,3].

At an international level, these institutions tend to be articulated around the governance of the territory, in relation to the capacity to make decisions that affect the political, public, and private framework of the city; planning around inter-level and cross-sectoral coordination of government; and management, through the provision of programs, policies and services, public and private investment, and organization according to local requirements [4]. In this sense, local governments and regional planners are configured as central actors of economic and social development [5,6].

Among the significant challenges of regional and local governments, particularly in Latin America, is the fact that concentration of wealth, income and employment, investment, and social mobility is highly asymmetrical in large cities, and very few are able to achieve significant improvements in terms of equity [5,7].

Lagging areas in Latin America face numerous public health challenges, including reduced access to health services, poor nutrition, uncontrolled cardiovascular disease, and mental disorders [8]. Understanding how progress varies by rurality and region can help identify urban or rural areas where progress is lagging, and improvement is needed, as they determine the public health priorities that are most relevant to different communities [9]. In this context, the Development of Territorial Plans must link knowledge to action to tackle the social determinants of health [10].

Territorial planning must respond to the local needs of the regions by means of development plans that incorporate different types of interventions—not limited to increasing incomes—that make it possible to reduce inequality by focusing resources on the most vulnerable and least developed areas, bearing in mind that these areas have the most unfavorable conditions for reducing poverty and that social results do not always follow financial results [11,12].

According to the international guidelines for urban and territorial planning, national governments, in coordination with other actors and levels of government, should establish the general stages, updates, monitoring and evaluation systems applicable to Territorial Development Plans (TDPs) [13]. Performance indicators and stakeholder participation should be fundamental to these systems [14]. In addition, developing TDPs should include multiple components, including a clear and feasible prioritization of expected results. Effective implementation and evaluation of TDPs require, in particular, continuous monitoring, periodic adjustments and adequate capacities at all levels [13].

According to the OECD, the evaluation of an ongoing or completed project, program, or policy is a systematic and objective assessment of its design, implementation, and results. The aim is to determine the relevance and achievement of the foals, as well as efficiency, effectiveness, impact, and sustainability for development. An evaluation should provide credible and valuable information to feed lessons learned into decision-making. Depending on the time for the assessment of social projects, different types of evaluation can be established, referring to the stages of Planning, Execution, Monitoring and Evaluation, and Reporting of Accounts or Results [2].

Although there is consensus on the need for planning, monitoring, and evaluation of TDPs, there are no well-defined tools on how each of these assessments should be conducted. Thus, several key questions arise, including how indicators of success should be measured—not only at the end of the period but from the time of initial planning and on a regular basis—and what tools could be introduced to evaluate these indicators. In general, it is recommended that these indicators be specific, concrete, and quantifiable and should be able to measure outcomes of different dimensions [11].

### 1.1. Chilean Context on Territorial Planning

Chile is one of the most centralized OECD countries [15]. Inequality in Chile has a strong regional component and a notable intra-territorial inequality [16]. According to the OECD territorial studies, it has been recommended to take a stronger regional approach to economic development to maximize regional opportunities and enhance the country’s development as a whole. There is a need to implement policies based on territory to improve the quality of regional investments and public services. To achieve these goals, the country requires greater involvement of institutional figures and regional actors in the planning and coordinating of the regional development plan [17].

With regard to territorial management, Chile has progressively adopted an innovative regional perspective through initiatives such as the creation of regional development agencies, a regional innovation fund and, more recently, the figure of regional governments and governors. Regional governments are the bodies responsible for the upper management in each region and are concerned with the region’s social, cultural, and economic development. The regional governor is the executive body of the regional government [18].

In terms of the assessment of the development of the territories, efforts have been made mainly in the design of indicators for measuring the territory in various dimensions. Among them is the Regional Development Index (IDERE, in Spanish), developed in 2016 as a tool to measure development at a territorial level from a multidimensional perspective [19]. Among its primary purposes is to provide data for public debate, generate new information that is useful for future studies and research, and supply key information for territorial planning [20].

Among the public policies aimed at reducing development gaps and promoting local development in different areas, Law 21.074 establishes the strengthening of the country’s regionalization by defining a National Policy on Lagging Areas [21]. The purpose of the National Policy on Socially Lagging Areas states the need *“to promote equal access to opportunities among people regardless of where they live, by focusing resources on those territories that reveal the widest gaps in their social development; to seek that these territories may reach levels that are not lower than those levels on their own region, through the coordinated work of public bodies and entities or actors of the private sector, present in the territory”*. This National Policy establishes the condition of isolation and the presence of social gaps as criteria for the identification of lagging areas: the condition of isolation understood as *“that condition faced by localities that have difficulties of accessibility and physical connectivity, have very low population density, present territorial distribution and inhabitants dispersion, and lack coverage of basic public services, according to the relationship between the structural isolation components and the degree of integration”*; and the presence of social gaps consisting of *“the distance between community poverty and regional poverty, which is understood as the difference between the average income poverty rate and the average multidimensional poverty rate of each community, and the regional average of both rates”* [22].

The stages included in this policy to consider the proposal of a given territory as a lagging area are, as indicated in Article 7 of Decree 975: an elaboration of the Territorial Development Plan (TDP) to propose territories as lagging areas, presentation of the proposal of territories as lagging areas and their respective Development Plan, evaluation of the proposed development plans and determination of a territory as a lagging area [22].

TDP is defined as a territorial planning tool consisting of a set of initiatives, actions and investments prioritized by the Regional Government for developing a given territory to overcome the lag in social matters. The elaboration of the TDP must contain, inter alia, a diagnosis, a definition of the baseline for a focused intervention to be carried out through the plan, the strategic and specific objectives of the plan, its goals and indicators of compliance, and a portfolio of publicly funded initiatives to be implemented to overcome or mitigate those factors that affect the consideration of the territory as a lagging area [22].

Regarding assessing the proposed development plans, the Undersecretariat of Regional and Administrative Development will evaluate compliance with the criteria and indicators established in the Regulations. Subsequently, the National Investment System must evaluate the portfolio of publicly funded initiatives proposed to overcome or mitigate gaps contained in the corresponding development plan [22].

### 1.2. Definition of the Problem and Objective of Study

As described above, evaluating the TDPs that meet the objectives of the public policies of various countries and that allows them to combine the different dimensions of territorial development is a challenge that most Latin American countries face, particularly in those territories that manifest an unfavorable position with respect to others. The problem arises when proposing a model to evaluate the TDPs that meet the criteria and indicators established in the framework of a National Policy or Regulation that is in accordance with local needs.

The problem of establishing a methodology to guide the evaluation emerges in Chile, as established in the National Policy on Socially Lagging Areas in Chile [22], which states that regional governments elaborate the TDPs, and the Undersecretariat of Regional Development carries out the evaluation.

Even though all territories are classified as lagging areas, not all have the same characteristics. In the territories, there are different needs and opportunities in various areas, the capacities of the financial resources to address those needs and options are vastly diverse, and there is generally a deficit in the human capital capacities required to draft a TDP. Moreover, a relevant aspect is that the focus or vision of the plan as defined by the Regional Government is different in each territory. Therefore, a methodology that allows for proposing an evaluation model to evaluate the TDP based on the items that the formulation must contain is required, regardless of the territory’s characteristics, the focus and areas of action defined. 

Considering the above, this paper addresses the following research questions:What dimensions or criteria should be considered in an evaluation model for TDPs that meets the requirements of the Policy?What methodological tool allows us to unify, in an objective way, the evaluation of the different TDPs, while managing to combine the different dimensions, objectives, scales and judgments of the experts present in the evaluation process without sacrificing the quality, reliability and participation of the other actors involved in the process?What elements should be considered in a TDP to ensure compliance with the focus or vision of territorial development in consistency with the characteristics of the territory?What quality criteria should be considered in formulating a TDP for its correct execution and continuity over time?

The objective of this research is to propose a model to evaluate TDPs under a multi-criteria methodology while presenting an application of this methodology for assessing TDPs for lagging areas in Chile, which guarantees compliance with what is required by the National Policy, and also ensuring, in its formulation, the consistency of quality criteria and development focus with the characteristics of the territory. 

This paper presents the methodology for developing the TDP evaluation model that emphasizes the advantages of using a multi-criteria approach, its results, and finally, a discussion and conclusions on the study’s main contributions and the model’s future lines of research. 

## 2. Materials and Methods

For the evaluation of the TDPs, in reference to compliance with the National Policy on social matters, the Analytic Hierarchy Process (AHP) multi-criteria methodology is applied [23]. The choice of this methodology is justified by the necessity to establish a method that combines the different dimensions, objectives, scales, and judgments of the experts present in the evaluation process without sacrificing the quality, reliability and participation of the different actors involved in the process.

The multicriteria methodology has been successfully implemented in different decision-making contexts such as selection, evaluation, cost-benefit analysis, allocation, planning and development, priority, and ranking, and in various disciplines such as health, economy, education, administration and finance, among others [24,25]. This methodology has also been implemented in terms of territorial planning with abundant practical applications with problems of management and allocation of water and soil [26]

However, although there are numerous applications in different disciplines, no research has been found that covers how to use the multi-criteria methodology to evaluate the formulation of Territorial Development Policies and Plans for less developed areas. That is the challenge of this research.

The formation of a panel of experts (decision makers) is required for the construction of the model. Experts can validate the model due to their experience and knowledge of the subject.

For this study, 16 experts were considered who represented different areas of the Undersecretariat of Regional and Administrative Development (SUBDERE, in Spanish) and participated in various stages pertaining to the three principles of the model development. This panel of experts is made up of professionals from the social field with higher education, a high level of experience (between 5 and 20 years in regional development programs), significant knowledge of lagging areas, and significant knowledge of the details of the Chilean National Policy; furthermore, six of them are executives with experience in decision-making.

A non-probabilistic sampling was used as criteria for the selection of the experts participating in the creation of the evaluation model, where knowledge and experience in TDP evaluation, knowledge of national policy, and work experience in lagging areas were considered.

A total of 14 workshops were held over three months: in the first two months, eight weekly workshops were held, lasting between 2 and 3 h; in the last month, six 3 h workshops were held. The workshops were organized according to the modeling stages: the creation of the model structure and its validation, assignment of priorities and logical consistency, and the creation of indicators and their validation.

All the workshops with the panel of experts were held in the SUBDERE offices, where the model criteria were discussed, and the hierarchical structure, indicators and measurement scales of the model were validated.

Furthermore, the axioms of the model were verified, and the priorities or weights of the hierarchical structure were established.

The methodology consists of three principles. The first principle concerns constructing a hierarchical structure, defined by the objective of the model, the strategic criteria, and the corresponding disaggregated sub-criteria for the terminal criteria. In addition, the construction of terminal criteria indicators and their respective measurement scales are considered; measurement scales being transformed into a common scale called the priority scale.

The second principle is the assignment of priorities or weights to the respective criteria that define the model. The assignment is not arbitrary but based on exact sciences, where pairwise comparison matrices are used, whereby experts enter judgments to a matrix by comparing two elements or criteria. The vector resulting from the pairwise comparison process relates to the priorities.

The third principle is the logical consistency of the judgments that experts enter into the matrices of pairwise comparisons. Consistency exists if transitivity of the judgments as well as proportionality, is met.

It is important to note that the four axioms of the methodology should be verified: Independence, Reciprocity, Homogeneity and Fulfillment of Expectations [23].

Two of the main characteristics of the multi-criteria methodology are the variety of factors that can be integrated into the evaluation process and that the evaluation result is synthesized in a common scale called the scale of priorities, of which values range from 0 to 1, allowing to provide a simple and clear interpretation. For example, a final evaluation result equal to 0.5 indicates that the evaluation achieves 50% of achievement compared to what the model requires. 

### Conceptual Model

Based on the AHP multi-criteria modeling, the evaluation model strategy is conceptually based on the principles that the National Policy requires to be fulfilled in formulating a TDP to bridge the gaps in lagging areas to improve the quality of life of the inhabitants of those territories. From this National Policy arises a series of foundations regarding understanding territories as lagging areas and the principles to be fulfilled when elaborating the TDPs for said lagging territories. 

Figure 1 summarizes the characteristics of the AHP multi-criteria modeling for the TDP evaluation of lagging areas.

## 3. Results

Below are the main results of the application of the methodology.

### 3.1. First Principle of Multi-Criteria Modeling: Creation of the Hierarchical Structure

The hierarchical structure of the model has been built considering the principles that the National Policy inspire [21], namely:Cross-sectoral action: Coordinated work among the different services and bodies of the State Administration, whose participation is relevant for the development of the said territory.Comprehensiveness: Social gaps under discussion must be analyzed in a multidimensional way to determine their causes and, thus, create or design targeted interventions to reduce them.Temporary nature of targeted interventions: The interventions will not be permanent but shall be valid pursuant to the TDP. The deadline will coincide with a territory no longer being a Lagging Area.Strengthening and development of local human capacities: Conditions for access to sustainable job opportunities should be created as well as vocational or technical training and policies that contribute to the retention of qualified human capital in the different communities and regions of the country.Public–private cooperation: Joint work among public bodies and entities or actors of the private sector present in the territory should be promoted to enhance targeted interventions to contribute to a territory no longer considered as a Lagging Zone in the shortest time and under the subsidiarity principle.Sustainability: The parties involved in the various stages of the process must tend towards overcoming social gaps and commit themselves through their actions.Transparency: Objective criteria will be applied to correctly measure the execution of resources in terms of the quality and relevance of the targeted interventions.

For the construction of the hierarchical structure, two of the research questions were discussed with the experts:What elements that ensure the focus of territorial development should be considered in a TDP?

The elements expected to be considered in a TDP are a comprehensive development vision that allows for the collection of the main potentialities and opportunities for development under cross-sectoral participation and a multidimensional analysis of both social gaps and opportunities and potentialities, so that a TDP with a Focus as a Value Bet is proposed. As such, a target vision of the TDP is realized with the design of relevant interventions aligned with the axes of territorial development to reduce social gaps.
2.What quality criteria should be considered in formulating a TDP for its correct execution and continuity over time?

The TDP is expected to consider the quality elements of the portfolio proposal, referring to the coherence of the portfolio of initiatives in the TDP with the objectives, work methodology and outcome indicators, feasibility of the portfolio regarding financing, and deadlines for its execution. On the other hand, it is expected to consider the principles of the sustainability of the initiatives and transparency of information.

The previous discussion was carried out in two of the workshops with the panel of experts, motivated by the two questions stated above through a debate methodology. The experts argued the elements and criteria that should be considered supported by what the National Policy requires.

Thus, the following hierarchical structure of the model was defined:

Objective of the model: Evaluate lagging areas in the TDP and ensure compliance with the National Policy.

#### Strategic Criteria of the Model


TDP focus conditions (target vision): It refers to the fulfillment of the conditions that ensure that the TDP is carried out with cross-sectoral participation and that it focuses on identifying and supporting the resources, potentialities, and opportunities of the territory as well as the problems that affect the territory under a multidimensional analysis of both the gaps and the opportunities and potentialities. These factors should be reflected in the proposal of a portfolio of initiatives relevant to the Focus of Value Bet of the territory and aligned to the axes of territorial development, to overcome social lag.TDP Quality Conditions: It refers to the quality conditions that ensure the execution of the plan. It includes the quality of the portfolio proposal and the quality of transversal principles.


The strategic criteria that emerge from the criterion, “Focus Conditions” (target vision of the TDP), from the model are shown in Figure 2.

The first strategic criterion, Comprehensiveness of the TDP, is divided into three criteria, which in turn are divided into sub-criteria, as can be seen in Figure 3.

The first criterion, “Comprehensiveness of the TDP,” refers to the global and comprehensive vision of the territory development, which identifies the resources, opportunities, and potentialities of the territory under cross-sectoral participation, as well as the problems associated with social gaps under a multidimensional analysis. The definitions of the “Integrality of the TDP” criterion sub-criteria are set out in Table 1.

The second criterion, “Alignment with the axes of territorial development”, refers to the level of alignment of the initiatives in the TDP with the axes or objectives of territorial development of lagging areas. In other words, the portfolio initiatives are consistent or in accordance with the axes of territorial development with reference to humans and capital, infrastructure conditions, working conditions and income increase (economic and productive development), and conditions to enhance opportunities and potentialities. These initiatives are associated with the dimensions that aim to reduce the social gaps in these territories and will be prioritized by the regional teams.

Meanwhile, the third criterion, “Relevance”, refers to the adequacy of the portfolio, which establishes the degree of relevance and suitability of the portfolio for the development of the territory, considering the reality in which it will be applied. In other words, it is measured if the TDP portfolio’s proposal of initiatives associated with the development axes are relevant, convenient, coherent, and/or consistent with the focus of the TDP and the planning tools. Adequate planning tools refer to whether the TDP is part of any of the guidelines established in the Regional Development Strategy and whether it considers other planning tools in force in the region and other sectoral planning tools involved in the territory in question.

The second strategic criterion, “Conditions of Quality of the Plan”, is divided into two sub-criteria: quality of the portfolio proposal and quality of cross-cutting principles. The sub-criteria for the Strategic Criterion “Quality Conditions of the Plan” are outlined in Figure 4. 

The first criterion, Quality of the Portfolio Proposal, refers, on the one hand, to the consistency or coherence of the TDP initiatives portfolio with the objectives, work methodology and outcome indicators of the TDP focus and, on the other hand, to the feasibility of the portfolio in relation to the financing and deadlines for its execution.

The definitions and forms of measurement of the sub-criteria for “Quality of the Portfolio Proposal” are shown in Table 2. 

The second criterion, “Transversal Quality Principles”, includes the principles of sustainability and transparency. Sustainability refers to the parties involved in the different stages of the process that must tend and commit themselves through their actions to overcome social gaps. Transparency is understood in relation to the accountability and dissemination of the TDP.

Within sustainability, the commitment of the regional government (GORE) is included; commitment to ensuring the development sustainability through sectoral, municipal, or private commitments to the maintenance and operation of the infrastructure generated. Commitment to participation and cooperation, according to the requirements of the national policy, is also regarded here, which includes public-private cooperation, that is, to promote joint work between public bodies and entities or actors of the private sector present in the territory, with the aim of promoting targeted interventions to contribute to the territory leaving the status of the lagging zone as soon as possible and under the principle of subsidiarity.

### 3.2. Second and Third Principles of the Model: Prioritization and Logical Consistency

In the priority-setting workshop, the team of actors or experts presented their judgments through the matrices of pairwise comparisons, arguing the assigned values and achieving consensus, in addition to verifying the consistency of the decisions. 

The AHP multi-criteria methodology was applied, supported by “Expert Choice” software, version 11 (Arlington, VA, USA). 

The local and global priorities for the model structure were obtained from the consensus work of the group of actors.

It is important to note that the local priority or Local weight (LW) refers to the weight obtained in the criteria of the same level, which must add 1 to the scale of proportions 0-1 or, in percentage, must add 100%. On the other hand, the global priority or global weight (GW) refers to the weight that a sub-criterion receives from the criterion before it, that is, from the “parent criterion”. In the case of level 1 criteria, local and global priorities are the same. Still, from level 2 and under, the global priorities are different, and the sum of the global priorities of the same level must match the global priority of their parent criterion. 

### 3.3. Summary of Overall Priorities of the Model Criteria

The strategic criteria “Conditions of the Focus of the Plan (target vision)” and “Conditions of Quality of the Plan” obtained equal weighting, that is, 50% priority for each one.

Figure 5 illustrates the priorities of the model criteria for each level. The sum of the priorities is 1. As can be seen, the quality of the portfolio proposal and the comprehensiveness of the plan are the criteria that add up to more than 50% of the importance in the evaluation of the plan as far as the formulation is concerned.

For each terminal criterion of the model, indicators were developed with their respective measurement scales, which the experts validated. 

The experts validated the model’s content, and the methodology’s four axioms were verified: independence, reciprocity, homogeneity, and fulfillment of expectations.

Once validated, the TDP evaluation tool was created based on the model. This tool also serves as a guide for the territorial teams to formulate a plan.

The model allows us to obtain a synthetic index for each territorial plan and aggregate indices for each criterion, allowing for the generation of priorities. Moreover, it will enable the weaknesses and strengths of each individual plan to be identified at an aggregate level, considering all the TDPs. The final synthetic index is calculated as an average of the terminal criteria values weighted by the global priorities of the terminal criteria. The overall priority of the criteria is shown in Figure 6.

Figure 6 shows the most important sub-criteria in the evaluation: “Feasibility of the Portfolio”; “Consistency”; “Comprehensive Vision of the Development of the Territory”; and “Sustainability”.

## 4. Discussion

The issue in the present study is how to assess a TDP formulation and apply the study to the evaluation of the plans formulated by the regional governments in Chile and accordance with the provisions of the National Policy on this country’s areas lagging behind in social matters [21].

According to the review of theoretical evidence and what other countries have put into practice, development plans are evaluated using different assessment guides. However, according to the knowledge of the authors, few publications account for how to face an evaluation of the formulation of a TDP in a comprehensive, objective way that captures elements of different levels of measurement.

An adequate evaluation of the plans is essential in that it shows “functions of continuous learning that improve the exercise of power and public management and create transparency to hold accountable the agents involved in the proposal of public policies, accountability, and communication. It is useful, among other things, for facilitating public interest decision-making, focusing attention on the objectives pursued, providing control mechanisms, and detecting and correcting errors in programs and projects” [27].

### 4.1. Strengths of the Model

In this research, the AHP multi-criteria methodology provided an evaluation model which includes a synthetic index for each TDP and aggregate indices for each criterion of the model. Dimensions are integrated into a single model where the requirements have different measurement scales. Those scales are transformed into a common scale.

The evaluation allows a better focus on the actions implemented to meet the TDP objective, thus helping to bridge the gaps in lagging areas in the territory. In addition, it facilitates the identification of the most relevant criteria for the fulfillment of the TDP objective. It also allows for detection, through the criteria and sub-criteria priorities, which indicators associated with them will be more relevant in the evaluation. Weights are assigned based on the exact science’s methodology, where the consistency of the judgments is checked, thus verifying compliance with content validity standards. Additionally, the evaluation model allows for detecting weaknesses and strengths when formulating the TDP. It favors the identification of weak aspects to be corrected before implementation. The formulation was sought to ensure the plan’s quality and compliance with the requirements of the Chilean National Policy. 

Furthermore, among the advantages offered by the proposal, the structure of the model for the evaluation of the development plan is carried out in a participatory manner, that is, considering the knowledge of experts and other professionals who participate in its formulation. 

Evaluators and formulators of the plans find, in the proposed model, clear and objective criteria and indicators, which can give greater credibility to the evaluation process.

### 4.2. Limitations and Future Research Lines

Although the model proposed here is not directly applicable to other countries, it serves as a reference example of a TDP evaluation methodology that can be used at an international level through modifications in the structure according to the specific requirements.

Different regional governments in Chile are currently applying the proposed model. It has been used initially to evaluate the plans of the Bío Bío, Coquimbo, and Maule regions, to subsequently implement the evaluation in Los Ríos, Ñuble, and Araucanía during the year 2023. Currently, the model is under study to be used in the plans of other Chilean regions.

As a future line of research, it would be interesting to compare the results of the evaluations of different development plans for other territories and analyze the knowledge retrieved from them over time. 

The methodology is applicable both for mid-term evaluations and subsequent evaluations to generate a system for assessing the TDPs that allows the territories to have a global and clear vision of the evaluation. 

## 5. Conclusions

The present study presents a methodology to evaluate the formulation of the TDP of lagging areas in Chile under an AHP multi-criteria methodology. This methodology allows us to objectively unify the evaluation of the different TDPs, and combine the different dimensions, objectives, scales, and judgments of the experts in the evaluation process without sacrificing the quality, reliability and participation of the different actors involved. The proposal not only responds to the need to evaluate the TDPs, but also supports the aspects contained in the formulation, regardless of the territory characteristics, the focus, or the scope of action defined.

Regarding the weights obtained through the multi-criteria methodology, equal importance was obtained for the two strategic criteria, that is, “Conditions of the Focus of the Plan (target vision)” and “Conditions of Quality of the Plan”. In the first level of the model structure, the sub-criteria that achieved 58% importance are “quality of the proposal” and “completeness of the plan”.

Thus, in the formulation of the plan, in order to reduce the social gaps associated with the problems identified in the focus of the plan, the adequate relation of the portfolio of initiatives with the objectives, the work methodology (referring to the coordination and monitoring of intervention strategies) and the result indicators (or goals) proposed for each initiative.

Next in importance is the “Integrality of the Plan”, which refers to the global and comprehensive vision of development of the territory, which identifies the resources, opportunities, and potentialities of the territory through intersectoral participation, as well as the problems associated with social gaps under a multidimensional analysis.

The methodology responds properly to the initial problems since it allows to establishment of a TDP evaluation considering the requirements demanded by the National Policy. The model is flexible to changes in the criteria and the priorities, being able to be updated according to the needs over time. Thus, it allows for feedback to the regions on the aspects that must be corrected or improved before implementing the initiatives of projects and programs proposed in the plan and also serves as a guide for the regions to formulate the plan correctly. Having clear criteria directly aligned to the National Policy and with objective indicators makes it easier for evaluators to conduct a reliable, objective, and transparent evaluation.

Likewise, the proposed methodology allows future monitoring of the plans, in those criteria of quality of the plan and, on the other hand, to obtain aggregated indices for each criterion or dimension of the model. An efficient and effective tool is provided to support decision-makers in the formulation of better TDPs that reduce gaps in lagging areas, having a very prominent role to play in improving the living conditions of groups of highly vulnerable people.

## Figures and Tables

**Figure 1 ijerph-19-11312-f001:**
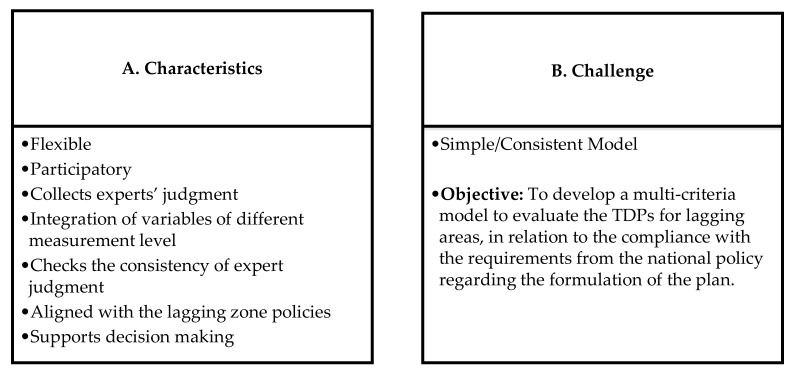
(**A**) Characteristics of the multi-criteria model to be developed for assessing TDPs for lagging areas; (**B**) challenge and objective of the multi-criteria model to be developed for the assessment of TDPs for lagging areas. TDP—Territorial Development Plans.

**Figure 2 ijerph-19-11312-f002:**
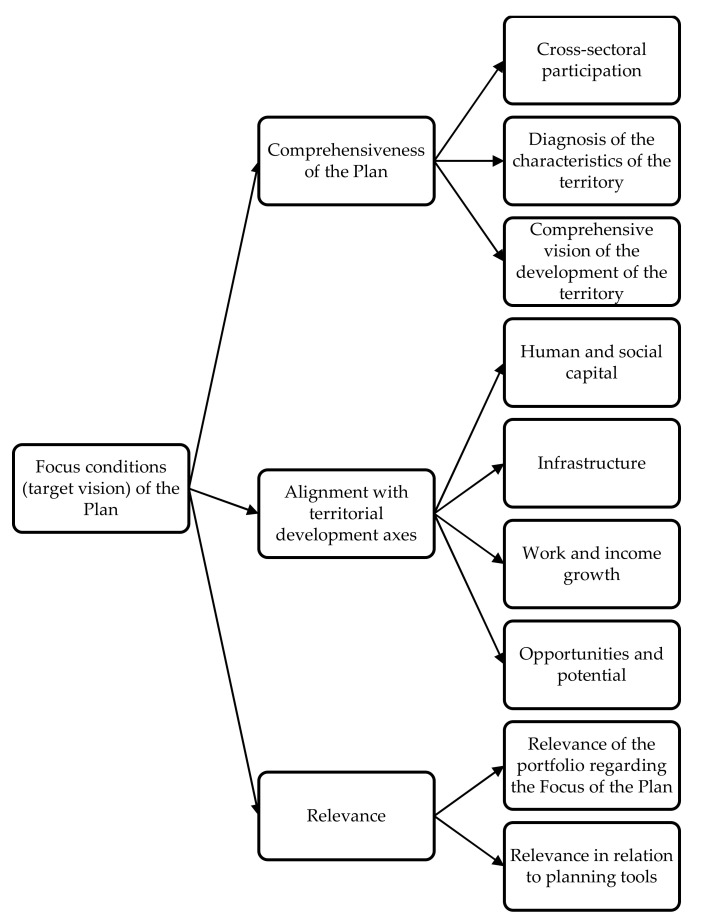
Criteria for the Strategic Criterion “TDP Focus Conditions”.

**Figure 3 ijerph-19-11312-f003:**
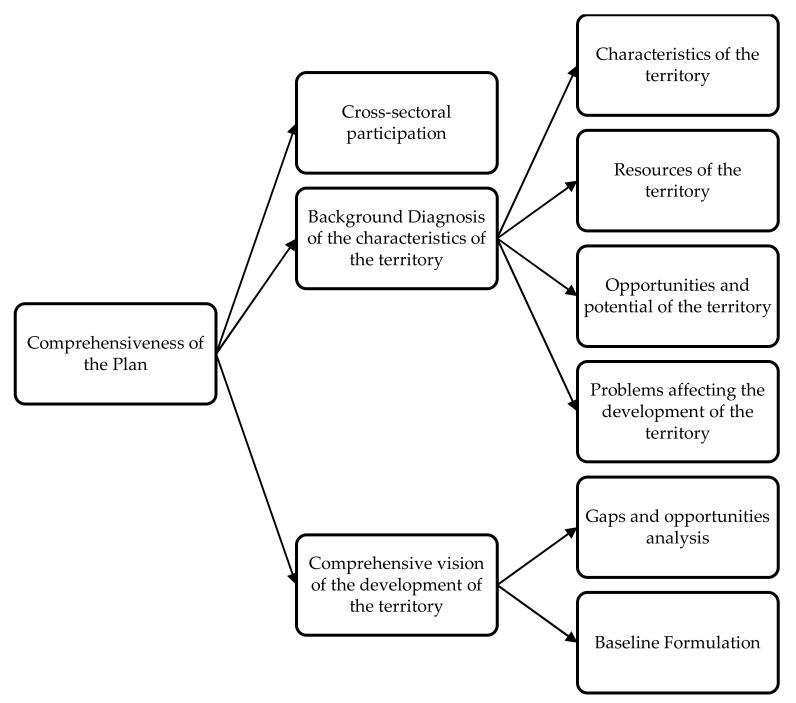
Sub-criteria for the Criterion “Comprehensiveness of the TDP”.

**Figure 4 ijerph-19-11312-f004:**
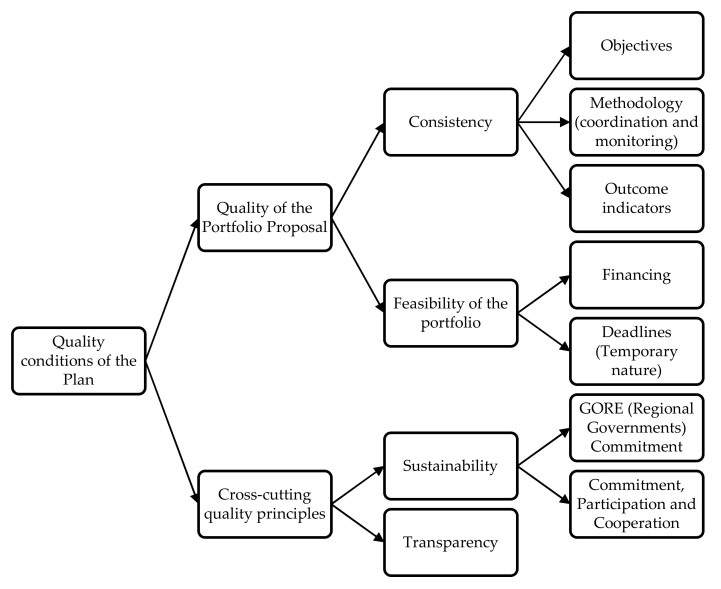
Criteria for the Strategic Criterion “TDP Quality Conditions”.

**Figure 5 ijerph-19-11312-f005:**
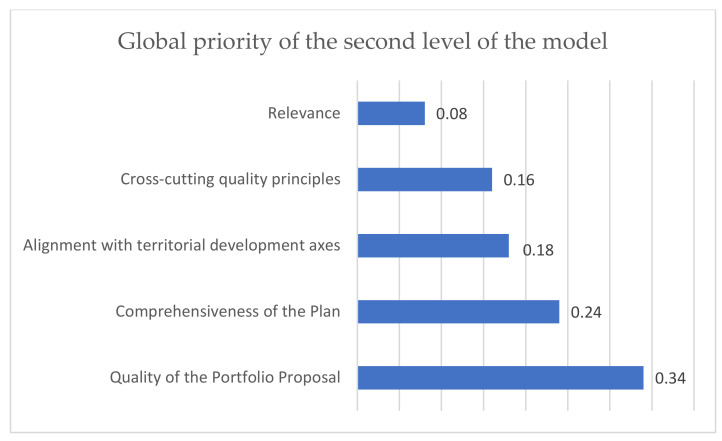
Priorities of the model criteria.

**Figure 6 ijerph-19-11312-f006:**
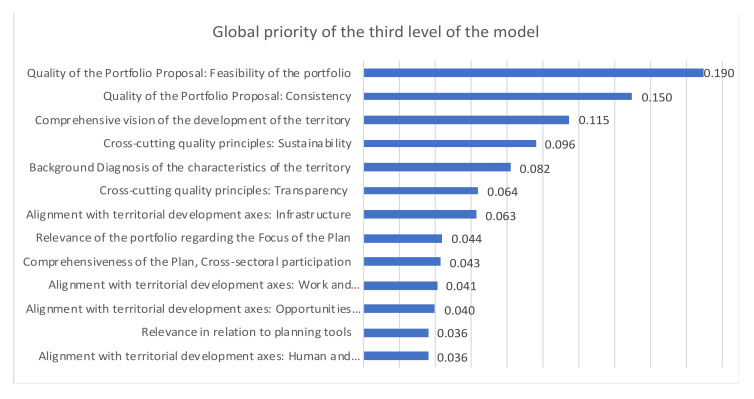
Overall priority of the criteria.

**Table 1 ijerph-19-11312-t001:** Definitions and forms of measurement of the sub-criteria for the Criterion “Comprehensiveness of the TDP”.

Sub-Criteria	Definition and Forms of Measurement
**Cross-sectoral Participation**	It is measured through the participation of different actors and sectors in working sessions to identify the needs and problems of the territory and to define and/or validate the focus of the plan as a value bet focus to help mitigate or overcome the lag in social matters. The criterion gathers the agreements from the participatory process in minutes and in a document that integrates and synthesizes the work done in the work sessions and whether it was known, discussed and validated by the participants.
**Diagnosis of the characteristics of the territory *^1^**	**Characteristics of the Territory:** Demography, availability of basic services, health and education services, housing conditions and environment, access to work and social security, territorial connectivity, access to telecommunications, access to public services, productive economic development and sustainable development.
	**Characteristics of the Resources of the Territory:** Human resources, social resources, physical resources, natural resources and heritage resources are considered.
	**Characteristics of the Opportunities and Potentialities of the territory:** Opportunities are the positive aspects that the territory can take advantage of using its strengths to benefit the territory. Potentialities refer to the capacities that territory can develop.
	**Identification and prioritization of the problems that affect the territory and, in particular, the focus of the plan.** Here are included a description of the problem, identification of the locality or localities that present the problem, description of the duration of the problem, recognition and description of the causes, and recognition and description of its effects.
**Comprehensive vision of the territory development *^2^**	**Gaps and opportunities analysis:** Refers to a multidimensional analysis that captures, on the one hand, the relationship between the prioritized development axes and the prioritized problems from which social gaps arise, and, on the other hand, an analysis of opportunities and potentialities to mitigate the gaps. This criterion allows us to gain a comprehensive vision of the territory.
	**Baseline Formulation:** The baseline allows us to understand the state of the art of the main variables that identify the development gaps of the territory associated with the problems identified in the focus of the plan. The baseline is understood as the first measurement of the variables or criteria contemplated in the plan. It shows if the plan identifies -for each prioritized problem- the aspects that a correct formulation of the initial state of the variables must contain. There will be a specific objective associated with the variables. The baseline serves as a reference point for further comparison to assess whether the objectives of the plan were achieved. The objectively verifiable Indicators of the Baseline shall be precise in terms of the characterization of the problems. They shall be clear, easily verifiable, not excessively numerous, and valid to be at the center of the project’s monitoring and mid-term and final evaluations. The Baseline shall establish reference values for the indicators of expected results.

Note: *^1^: It is measured by means of four criteria: Characteristics of the Territory, Characteristics of the Resources of the Territory, Characteristics of the Opportunities and Potentialities of the Territory, and Identification and Prioritization of Problems. *^2^: It refers to a comprehensive perspective of the territory, which presents a multidimensional analysis of social gaps, opportunities and potentialities of the territory and the correct formulation of the baseline.

**Table 2 ijerph-19-11312-t002:** Definitions and forms of measurement of the sub-criteria for the Sub-criterion “Quality of the portfolio proposal”.

Sub-Criterion	Definition and Forms of Measurement
**Consistency *^1^**	**Objectives:** To measure the level of consistency with the development factor or axis and the focus of the plan is measured for each strategic objective (associated with the development factor or axis), and the level of consistency with the strategic objective and the variables identified in the baseline associated with a given problem is measured for each specific objective.
	**Work methodology:** It is evaluated whether the work methodology for the execution of the plan contemplates periodic systematic coordination and monitoring, clearly defining the role of the actors in charge of each stage or process in the execution of the initiatives from the plan.
	**Outcome indicators:** Indicator: It is the relationship (as a quotient) between two or more variables that allow us to verify the level of achievement attained in the fulfillment of the objectives. The plan must have a set of indicators that provides quantitative information regarding the achievement or result pertaining to each specific objective.
**Portfolio feasibility *^2^**	**Financing Feasibility:** For each initiative in the plan, the type and amount of financing are considered: provision, regional governments (GORE), sectoral, and total estimate.
	**Deadlines Feasibility:** The execution period and planning time are considered for each initiative.

Note: *^1^ It refers to the appropriate relationship or link of the portfolio of initiatives with the objectives, work methodology (coordination and monitoring of the strategies of the plan intervention), and the outcome indicators or goals proposed for each initiative to reduce the social gaps associated with the problems outlined in the TDP focus. *^2^ It refers to the feasibility in terms of the type of financing and the temporary nature regarding execution deadlines.

## Data Availability

Not applicable.

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
