# Peer review of "Proposal of a Multi-Criteria Model for the Evaluation of Territorial Development Plans: An Application in Chile’s Lagging Areas"

_ijerph, 2022, doi:10.3390/ijerph191811312_

Round 1
Reviewer 1 Report
The study is based on a detailed research and analyzes the problem of territorial planning with a consistent methodology. The content of the title is adequate, but the repetition of the word "development" is a bit confusing, it is recommended to replace it with a synonym in the first case. The introduction of the study, the indication of the subject of the research and the development of the context are appropriate. The discussion provide a well-structured, detailed overview, although the conclusion is somewhat rough and too general in comparison. It is recommended to revise the conclusion and formulate the conclusions more precisely. If possible, it may be worthwhile to improve the graphics of the figures and diagrams in order to convey the structures with more expressive means.
Author Response
The study is based on a detailed research and analyzes the problem of territorial planning with a consistent methodology.
1.- The content of the title is adequate, but the repetition of the word "development" is a bit confusing, it is recommended to replace it with a synonym in the first case.
2.- The introduction of the study, the indication of the subject of the research and the development of the context are appropriate. The discussion provides a well-structured, detailed overview, although the conclusion is somewhat rough and too general in comparison. It is recommended to revise the conclusion and formulate the conclusions more precisely.
3.- If possible, it may be worthwhile to improve the graphics of the figures and diagrams in order to convey the structures with more expressive means.
Response to reviewer 1:
1.- The title has been modified according to reviewer’ suggestion: “Proposal of a Multi-criteria Model for the Evaluation of Territorial Development Plans: An Application in Chile’s Lagging Areas”.
2.- The conclusion section has been modified according to reviewer’ suggestion: “In the present study, a methodology is presented to evaluate the formulation of the TDP of lagging areas in Chile, under an AHP multi-criteria methodology. This meth-odology allows to unify -in an objective way- the evaluation of the different TDPs, managing to combine the different dimensions, objectives, scales, and judgments of the experts who are present in the evaluation process. All of this, without sacrificing the quality, reliability and participation of the different actors involved in the process. The proposal not only responds to the need to evaluate the TDPs, but also supports the aspects contained in the formulation, regardless of the territory characteristics, the focus and the scope of action defined.
Regarding the weights obtained through the multi-criteria methodology, equal importance is obtained for the two strategic criteria, that is, for "Conditions of the focus of the Plan (target vision)" and "Conditions of Quality of the Plan". It is noteworthy that in the first level of the model structure, the sub-criteria that achieve 58% of importance are "quality of the proposal" and "completeness of the plan".
Thus, in the formulation of the Plan, in order to reduce the social gaps associated with the problems identified in the focus of the Plan, the adequate relation of the portfolio of initiatives with the objectives, the work methodology (referring to the coordination and monitoring of intervention strategies) and the result indicators (or goals) proposed for each initiative.
Next in importance is the "Integrality of the Plan", which refers to the global and comprehensive vision of development of the territory, which identifies, through inter-sectoral participation, the resources, opportunities, potentialities of the territory, as well as the problems associated with social gaps under a multidimensional analysis.
The methodology responds properly to the initial problems, since it allows to es-tablish a TDP evaluation considering the requirements demanded by the National Policy. The model is flexible to changes in the criteria and the priorities, being able to be updated according to the needs over time. So, it allows feedback to the regions on the aspects that must be corrected or improved before implementing the initiatives of projects and pro-grams proposed in the Plan and also serves as a guide for the regions to formulate the Plan correctly. Having clear criteria, directly aligned to the National Policy and with objective indicators, makes it easier for evaluators to carry out a reliable, objective, and transparent evaluation.
Likewise, the proposed methodology allows future monitoring of the plans, in those criteria of quality of the plan and on the other hand to obtain aggregated indices for each criterion or dimension of the model. An efficient and effective tool is provided to support decision makers in the formulation of better TDPs that reduce gaps in lagging areas, having a very prominent role to play in improving living conditions of groups of highly vulnerable people.”
3.- The figures have been modified according to reviewer’ suggestion: (figures 5 and 6).

Reviewer 2 Report
Dear Authors,
I want to thank for your submission and the opportunity to review an interesting manuscript focused on a proposal of a model of evaluation of lagging areas. The model can be used for the evaluation of territorial development plans in order to identify the priorities of development on different aspects (social, economic, cultural, environmental). The model of evaluation is built for lagging areas in Chile and can be adapted in order to be applied also for other countries located in Latin America or in other continents. The model is based on a methodology that facilitates an objective evaluation of different plans; it includes a combination of different dimensions, objectives, scales, and judgments of the experts included in the valuation process in order to contribute through their qualitative expertise to the development of the lagging areas.
Introduction
You can include one-two sentences about the importance of public health for the development of lagging areas. There is no mentioned about this aspect, especially since the manuscript is sent for the evaluation to the Vulnerable Communities and Public Health Special Issue.
Methodology: Has the Analityc Hierarcy Proces (AHP) multi-criteria methodology been applied in relation with the topic approached by you in several studies focused on different countries ? You used only one citation [21]. If yes, you should mention several examples. You can offer a synthetic explanation about the degree of efficienty of the application of this multi-criteria methodology in developed countries and in less developed ones.
Can you mention the experise of 16 experts selected by you?
You mentioned that you discussed two of the research questions with experts. Can you specify whether the discussions were organized as a semi-structured intervierws?
You also mention about the 14 workshops that were held over three months. Can you offer more information about them: when they are organized? their duration (one day, several days?), were they divided in several topics? How they were organized? Were they hosted by public institutions or in academic institutions?
4.2. Limitations and Future Research lines
You mention that the proposed model is currently being applied by different regional governments in Chile (Lines 452-453). You should mention these examples.
Author Response
I want to thank for your submission and the opportunity to review an interesting manuscript focused on a proposal of a model of evaluation of lagging areas. The model can be used for the evaluation of territorial development plans in order to identify the priorities of development on different aspects (social, economic, cultural, environmental). The model of evaluation is built for lagging areas in Chile and can be adapted in order to be applied also for other countries located in Latin America or in other continents. The model is based on a methodology that facilitates an objective evaluation of different plans; it includes a combination of different dimensions, objectives, scales, and judgments of the experts included in the valuation process in order to contribute through their qualitative expertise to the development of the lagging areas.
1.- Introduction: You can include one-two sentences about the importance of public health for the development of lagging areas. There is no mentioned about this aspect, especially since the manuscript is sent for the evaluation to the Vulnerable Communities and Public Health Special Issue.
2.- Methodology: Has the Analityc Hierarcy Proces (AHP) multi-criteria methodology been applied in relation with the topic approached by you in several studies focused on different countries? You used only one citation [21]. If yes, you should mention several examples. You can offer a synthetic explanation about the degree of efficiently of the application of this multi-criteria methodology in developed countries and in less developed ones.
3.- Can you mention the expertise of 16 experts selected by you?
4.- You mentioned that you discussed two of the research questions with experts. Can you specify whether the discussions were organized as a semi-structured intervierws?
5.- You also mention about the 14 workshops that were held over three months. Can you offer more information about them: when they are organized? their duration (one day, several days?), were they divided in several topics? How were they organized? Were they hosted by public institutions or in academic institutions?
6.- Limitations and Future Research lines: You mention that the proposed model is currently being applied by different regional governments in Chile (Lines 452-453). You should mention these examples.
Responses to reviewer 2:
1.- A paragraph has been included in the introduction section, according to reviewer’ suggestion: “Lagging areas in Latin America faces numerous public health challenges, including reduced access to health services, poor nutrition, uncontrolled cardiovascular disease, and mental disorders [8]. Understanding how progress varies by rurality and region can help identify urban or rural areas where progress is lagging, and improvement is needed, as they determine the public health priorities that are most relevant to different communities [9]. In this context, the Development of Territorial Plans must link knowledge to action in tackling social determinants of health [10]”. Furthermore, we have included three new references and updated the numbering.
2.- Two paragraphs have been included in the Material and Methods section, according to reviewer’ suggestion: “The multicriteria methodology has been successfully implemented in different decision-making contexts such as: selection, evaluation, cost-benefit analysis, allocation, planning and development, priority and ranking, and in various disciplines such as: health, economy, education, administration and finance, among others [25, 26]; also, in territorial planning, with abundant practical applications with problems of management and allocation of water and soil [27]
However, although there are numerous applications in different disciplines, no research has been found that covers how to use the multi-criteria methodology to evaluate the formulation of Territorial Development Policies and Plans for backward areas. That is the challenge in this research”. Furthermore, we have included three new references and updated the numbering.
3.- A paragraph has been included in the introduction section, according to reviewer’ suggestion: “This panel of experts is made up of professionals from the social field, with higher education, a high level of experience (between 5 and 20 years in regional development programs), great knowledge of lagging areas, and with great knowledge of the details of the Chilean National Policy; furthermore, six of them are executives with experience in decision-making”.
4.- A paragraph has been included in the Results section, according to reviewer’ suggestion: “The previous discussion was carried out in two of the workshops with the panel of experts, motivated by the two questions stated above, through a debate methodology, in which the experts argued the elements and criteria that should be considered supported by what National Policy requires”.
5.- Two paragraphs have been included in the Materials and Methoss section, according to reviewer’ suggestion: “14 workshops were held over three months: in the first two months, 8 weekly workshops were held, lasting between 2 and 3 hours; in the last month, six 3-hour workshops were held. The workshops were organized according to the modeling stages: creation of the model structure and its validation, assignment of priorities and logical consistency, creation of indicators and their validation.
All the workshops with the panel of experts were held in SUBDERE offices, where model criteria were discussed and the hierarchical structure, indicators and measurement scales of the model were validated”.
6.- A paragraph has been included in the Limitations and Future Research Lines section, according to reviewer’ suggestion: “It has been used initially to evaluate the Plans of the Bío Bío, Coquimbo, and Maule regions, to subsequently implement the evaluation in Los Ríos, Ñuble, and Araucanía during the year 2023. Currently, the model is under study to start using it in the Plans of other Chilean regions”.
